# Estimation of Olfactory Sensitivity Using a Bayesian Adaptive Method

**DOI:** 10.3390/nu11061278

**Published:** 2019-06-05

**Authors:** Richard Höchenberger, Kathrin Ohla

**Affiliations:** 1Institute of Neuroscience and Medicine INM-3, Research Center Jülich, 52428 Jülich, Germany; richard.hoechenberger@gmail.com; 2Psychophysiology of Food Perception, German Institute of Human Nutrition Potsdam-Rehbrücke, 14558 Nuthetal, Germany

**Keywords:** smell sensitivity, olfaction, threshold, staircase, QUEST

## Abstract

The ability to smell is crucial for most species as it enables the detection of environmental threats like smoke, fosters social interactions, and contributes to the sensory evaluation of food and eating behavior. The high prevalence of smell disturbances throughout the life span calls for a continuous effort to improve tools for quick and reliable assessment of olfactory function. Odor-dispensing pens, called Sniffin’ Sticks, are an established method to deliver olfactory stimuli during diagnostic evaluation. We tested the suitability of a Bayesian adaptive algorithm (QUEST) to estimate olfactory sensitivity using Sniffin’ Sticks by comparing QUEST sensitivity thresholds with those obtained using a procedure based on an established standard staircase protocol. Thresholds were measured twice with both procedures in two sessions (Test and Retest). Overall, both procedures exhibited considerable overlap, with QUEST displaying slightly higher test-retest correlations, less variability between measurements, and reduced testing duration. Notably, participants were more frequently presented with the highest concentration during QUEST, which may foster adaptation and habituation effects. We conclude that further research is required to better understand and optimize the procedure for assessment of olfactory performance.

## 1. Introduction

The appreciation of food involves all senses: sight, smell, taste, touch, and also hearing. While the sight of a cup of coffee may indicate its availability, it is typically its smell that makes it appealing and that triggers an appetite for most people. During consumption, the smell or aroma is perceived again retronasally and supported by its pleasant temperature and a bitter taste. These largely parallel sensations occur automatically and only raise awareness when one or more senses are disturbed. That said, the sense of smell has been shown to influence food choice and eating behavior [1], and its impairment has even been associated with a higher risk for diet-related diseases like diabetes [2]. Even more, olfactory stimuli can invoke emotional states, are linked to memory storage and retrieval, and as such also serve as important cues to rapid detection of potentially dangerous situations and threats (see e.g., [3,4]. Given that the estimated prevalence of smell impairment is 3.5% in the United States [5], continuous efforts are made toward an efficient and precise assessment of olfactory function.

The Sniffin’ Sticks test suite (Burghart, Wedel, Germany; [6]), is an established tool in the assessment of olfactory function. It consists of three tests involving sets of impregnated felt-tip pens: odor detection threshold (T), odor discrimination (D), and odor identification (I). Each test produces numbers in the range from 1 to 16 (T) or from 0 to 16 (D and I) as a performance measure. Overall olfactory function is assessed by summing all three test results, resulting in the *TDI score.* Comparison of individual TDI scores to the comprehensive set of available normative data (e.g., [7,8]) facilitates the interpretation of test scores and allows to reliably diagnose olfactory impairment. Notably, threshold, discrimination, and identification measure different facets of olfactory function [9]. The threshold, however, has been found to explain a larger portion of variability in TDI scores than the two other measures [10]. Moreover, the discrimination and identification tests follow relatively simple test protocols in which all stimuli are presented only once and in a predefined order. The threshold, in comparison, is of a more complex nature, and the method, therefore provides the largest potential for possible improvements. It follows a so-called adaptive method, specifically, a “transformed” one-up/two-down staircase procedure [11]. The procedure first assesses a starting concentration and then moves on to the “actual” threshold estimation, during which fixed step widths are used: for each incorrect answer, the stimulus concentration is increased by one step; and for two consecutive correct answers, the stimulus concentration is decreased by one step [6].

Since the one-up/two-down staircase was first conceived, several new approaches to threshold estimation, including Bayesian methods, have been published. Bayesian methods estimate parameters of the psychometric function (e.g., threshold or slope) using Bayesian inference: based on prior assumptions about the true parameter value, the stimulus concentration to be presented next is selected such that the expected information gain (about the parameter) is maximized. The first published Bayesian adaptive psychometric method is the QUEST procedure [12], which is still popular today. QUEST has two distinct properties that set it apart from the staircase described above. Firstly, it always considers the entire response history and is not solely based on the past one or two trials to select the optimal stimulus concentration to be presented next. Secondly, QUEST is not tied to a fixed step width, allowing it to traverse through a large range of concentrations more quickly.

In a clinical setting, at the otorhinolaryngologist’s (ear-nose-throat, ENT) practice or at the bedside in the hospital, shorter testing times are always beneficial, as they reduce strain on patients and free up time for other parts of diagnostics and treatment. But also when working with healthy participants, e.g., in a psychophysical lab or in large cohort studies, reduced testing time spares resources and allows for a larger number of measurements in a given time.

QUEST has been shown to converge reliably and quickly in gustatory threshold estimations [13,14]. Inspired by these results we set out to design and test a QUEST-based procedure for olfactory threshold estimation and to compare its performance with that of the established staircase method.

## 2. Materials and Methods

### 2.1. Participants

36 participants (32 women; median age: 29.5 years, age range: 19–61 years) completed the study. The influence of gender on olfactory performance has been investigated in previous studies. The results typically showed no (e.g., [15], several hundred participants; [7], >3000 participants, no main effect) or only rather small gender differences with negligible diagnostic and real-world relevance (e.g., [8], >9000 participants). We therefore did not enforce a gender balance in our sample. Due to a technical error, the identification test data was not recorded for one participant (female, 26 years old). All participants were non-smokers and reported being healthy and not having suffered from an infectious rhinitis for at least two weeks before testing. The study conformed to the revised Declaration of Helsinki and was approved by the ethical board of the German Society of Psychology (DGPs).

### 2.2. Stimuli

Stimuli were so-called Sniffin’ Sticks (Burghart, Wedel, Germany; [6]), felt-tip pens filled with an odorant. The Sniffin’ Sticks test battery consists of three subtests: an odor threshold test, an odor detection test, and an odor identification test. The threshold test comprises 48 pens. There were 16 pens filled with different concentrations of 2-phenylethanol (rose-like smell) ranging from 4% to approx. $1.22\;\times 10^{-4}\%$ (a geometric sequence with the common ratio of 2, so the first pen contained a 4% dilution, the second 2%, the third 1%, and so on), dissolved in 4% propylene glycol, an odorless solvent. Note that in this test, the 1st pen contained the highest, the 16th pen the lowest odorant concentration. The remaining 32 pens contained 4% propylene glycol and served as blanks. The pens were arranged in triplets such that each triplet contained one pen with odorant and two blanks. The detection test comprised 48 pens that were filled with 16 different odorants at supra-threshold concentrations. The pens were arranged in triplets such that two pens contained the same and one pen a different odorant. The identification test comprised 16 pens filled with different odorants at supra-threshold concentrations.

### 2.3. Procedure

#### 2.3.1. Experimental Sessions

Participants were invited for two experimental sessions – the Test and Retest session for the odor threshold. To ensure similar testing conditions across sessions, participants were instructed to refrain from eating and drinking anything but water 30 min before visiting the laboratory. Further, both sessions were scheduled at approximately the same time of day, and took place with a median inter-session interval of 3.0 days (SD = 2.6, range: 0.9–8.9 days); only four participants had an inter-session interval of more than 7.0 days. In each session, olfactory detection thresholds were determined using two distinct algorithms, staircase and QUEST, described below. The order of algorithms was balanced across participants and kept constant for Test and Retest within each participant. Additionally, odor discrimination and odor identification ability were measured at the end of one session following the standard Sniffin’ Sticks protocol (Burghart, Wedel, Germany).

#### 2.3.2. Stimulus Presentation

Testing took place in a well-ventilated testing room and was performed by the same experimenter, who refrained from using any fragrant products (e.g., soap, lotion, perfume, etc.) and wore odorless cotton gloves when presenting the stimuli. At the beginning of each test session, participants were blindfolded. To present a stimulus, the experimenter removed the cap from the pen, held the tip of the pen in front of the participant’s nose, approx. 2 cm from the nostrils, and asked the participant to take a sniff. For the threshold test, participants were blindfolded and informed that the odorant may be presented in very low concentrations, and that only one of the three pens presented in each trial contained the odorant, while the others contained the solvent exclusively. The task was to “indicate which of the three pens smells different from the others”, and participants had to provide a response even when unsure. Participants were familiarized with the odorant by presenting pen no. 1 (highest concentration) before testing commenced.

A similar procedure was used for the discrimination test: participants were blindfolded and presented with a triplet of pens containing clearly perceivable odorants. Each triplet consisted of two pens with the same and one pen with a different odorant. Again, participants were to indicate the pen that smelled different from the others. During threshold and discrimination testing, stimulus triplets were presented during each trial, which lasted approx. 30 s and included the presentation of three pens (approx. 3 s each) and a pause of 20 s. These tests yield a probability of 13 of guessing correctly.

For the identification task, the blindfold was removed and participants smelled one pen at a time. They were to identify the odor by pointing to the matching word on a response sheet with four written response options. The interval between pens was approx. 30 s. The probability of guessing correctly in this task was 14.

#### 2.3.3. Staircase

Following the standard protocol as detailed in the test manual; see also [16]), the order of presentation within the triplets varied from trial to trial. In the first trial, the odor pen was presented first, in the second trial, it was presented between two blanks, and in the third, after two blanks. After the third trial, this sequence was repeated.

We first determined the starting concentration. Beginning with the presentation of triplet no. 16 or 15 (balanced across participants), participants had to indicate which of the pens smelled different. Concentration was increased in steps of two (e.g., from pen 16 to 14) for each incorrect response. Once participants provided a correct response, the same triplet was presented again. If the response was incorrect, the concentration was increased again by two steps as before. However, if the triplet was correctly identified a second time, that dilution step served as the starting concentration.

Contrary to the standard protocol, where testing would then continue without interruption, our participants were granted a short break of approx. 1 min before the actual threshold estimation started with the presentation of the triplet containing the starting concentration. The threshold was determined in a one-up/two-down staircase procedure: odor concentration was increased by one step after each incorrect response (one-up), and decreased by one step after two consecutive correct responses at the same concentration (two-down). This kind of staircase targets a threshold of 70.71% correct responses ([11]; but cf. [17], who found small deviations from this value). That is, if presented repeatedly with a stimulus at threshold intensity, participants would be able to correctly identify it in about 71 out of 100 cases. The probability of providing *two consecutive* correct responses purely by guessing is 13 × 13 = 19. The procedure finished after seven reversal points were reached. The final threshold estimate was the mean of the last four reversal concentrations. This procedure is referred to simply as staircase throughout the this manuscript.

#### 2.3.4. QUEST

QUEST requires to set parameters that describe the assumed psychometric function linking stimulus intensity and expected response behavior. We assumed a sigmoid psychometric function of the Weibull family, as proposed by [12] (albeit in a slightly different parametrization) and used for gustatory testing [13], with a slope β=3.5, a lower asymptote γ=13 (chance of a correct response just by guessing), and a parameter λ=0.01 to account for lapses (response errors due to momentary fluctuation of attention):$$\Psi \left(x\right)=\lambda \gamma +\left(1-\lambda \right)[1-\left(1-\gamma \right)\exp\left(-10^{\beta (x+T)}\right)]$$

Here, the presented concentration is denoted as *x*, and the assumed threshold as *T*. This yielded a function extending from 0.33 to 0.99 in units of “proportion of correct responses”. The granularity of the concentration grid was set to 0.01. All parameters of this function were constant, except for the threshold, which was the parameter of interest that was going to be estimated in the course of the procedure. The prior estimate of the threshold was a normal distribution with a standard deviation of 20, which was centered on the concentration of pen no. 7, which was used as the starting concentration. The algorithm was set to target the threshold at 80% correct responses, which is slightly higher than the threshold target in the staircase procedure, but had proven to produce good results both in pilot testing as well as in gustatory threshold estimation [13,14]. Unlike in the staircase procedure, where the order of pen presentation varied systematically from triplet to triplet, triplets were presented in random order during the QUEST procedure.

Notably, QUEST updates its knowledge on the expected threshold after each response and proposes the concentration to present in the next trial such that it maximizes the expected information gain about the “true” threshold. As the set of concentrations was discrete and limited to 16, QUEST might propose concentrations other than those contained in the test set. In this case, the software selects the triplet with the concentration closest to the one proposed. In contrast to the staircase, where the concentration was always decreased or increased by a single step after the starting concentration had been determined, the step width was not fixed in QUEST. For example, QUEST might step up three concentrations in one trial, step down two in the next, and present the exact same concentration again in the following trial. Whenever the same concentration had been presented on two consecutive trials, the concentration for the next trial was decreased if both responses were correct, and increased if both responses were incorrect. QUEST might suggest to present concentrations outside of the range of available dilution steps. Therefore we set up the algorithm such that, whenever the presentation of a pen below 1 or above 16 was suggested, we would instead present pen no. 1 and 16, respectively. QUEST would be informed about the actually presented pen, and incorporate this information into the threshold estimate. Note, however, that final threshold estimates outside the concentration range could still occur occasionally, and needed to be dealt with accordingly; see the data cleaning paragraph in the next section for details.

The procedure ended after 20 trials. The final threshold estimate is the mean of the posterior probability density function of the threshold parameter. We will refer to this procedure as “QUEST”.

#### 2.3.5. Analysis

##### Odor Discrimination and Identification

The discrimination and identification tests comprised 16 trials each. For each test, the number of correct responses was summed up, resulting in a test score which can range from 0 to 16. Together with the staircase threshold, which yielded values from 1 to 16, the sum of all three test results formed a cumulative score: the TDI score.

##### Data Cleaning

When a participant reached one of the most extreme concentrations (i.e., pens no. 1 or 16) and provided a response that would, theoretically, require us to present a concentration outside the stimulus of set, the staircase procedure cannot be safely assumed to yield a reliable threshold estimate anymore. For example, if a participant fails to identify the highest concentration (pen no. 1), the staircase procedure would accordingly demand to present a hypothetical pen no. 0, which obviously does not exist. Since our sole termination criterion was “seven reversals”, we would repeatedly present pen no. 1 until a correct identification allowed the procedure to move up to pen no. 2 again. The resulting threshold estimate, then, would systematically overestimate this participant’s sensitivity. Therefore we set the threshold values of staircase runs where participants could not identify pen no. 1 at least once to T=1 after the run was completed, following [7] (but cf. [16], who suggest to set the value to T=0 instead). This was the case in five out of the 72 staircase threshold measurements (two during test, three during retest; five participants affected). Conversely, when a participant were to correctly identify the lowest concentration (pen no. 16), the staircase procedure would require the presentation of a hypothetical pen no. 17, in which case we would have assigned a threshold value of T=16; however, this situation did not occur in the present study after the starting concentration had been determined.

For QUEST, pen no. 1 was not correctly identified at least once in 12 of the 72 measurements, concerning 11 participants; no participant reached and correctly identified pen no. 16. QUEST yielded final threshold estimates T<1 in 11 measurements (8 during Test, 3 during Retest; 10 participants affected). Similarly to the data cleaning procedure for the staircase, we assigned threshold T=1 in these cases. Notably, this again concerned 3 of the 5 participants for whom we had assigned T=1 in a staircase experiment.

##### Test–Restest Reliability

To establish test–retest reliability, we first compared the means of Test and Retest thresholds for each procedure. Q–Q plots and Shapiro–Wilk tests revealed that thresholds were not normally distributed for the QUEST test session (W=0.90, p<0.01); we, therefore, compared the means using non-parametric Wilcoxon signed-rank tests. We then correlated Test and Retest threshold estimates via Spearman’s rank correlation (Spearman’s rho, denoted as ρ) to estimate the degree of monotonic relationship between measurements. Ordinary least squares (OLS) models were used to fit regression lines to provide a better understanding of the nature of the relationship between the threshold estimates (i.e., whether test thresholds could predict retest thresholds). Q–Q plots and Shapiro–Wilk tests showed that the regression residuals were normally distributed (all p>0.05) and thus satisfied an important requirement for OLS regression.

Although correlation and regression analyses are widely used to assess test–retest reliability and to compare methods, it has been argued that these measures may in fact be inappropriate (see e.g., [18,19,20]). Instead, analyses that focus on the *differences* between, not agreement of, measurements should be preferred. A possible approach is to calculate the mean difference $\bar{d}$ and standard deviation of the differences between two measurements to derive *limits of agreement,*$\bar{d}\pm 1.96\times \mathrm{SD}$ [18]. These limits correspond to the 95% confidence interval. This means that in 95 out of 100 comparisons, the difference between two measurements can be expected to fall into this range. Narrower limits of agreement indicate a better agreement between two measurements. The related repeatability coefficient (RC) was simply 1.96×SD, and its interpretation was very similar to the limits of agreement: only 5% of absolute measurement differences will exceed this value, and a smaller RC indicates better agreement. (It should be noted that an alternative method for calculating the repeatability coefficient has been suggested, based on the within-participant standard deviation, $s_{w}$ [20]. The results we obtained from these calculations were similar to those based on the standard deviation of the measurement differences. Because the latter are directly visualized in the Bland–Altman plot by the limits of agreement, i.e., meandifference±1.96×SD, we opted to only report these values.)

If the differences between two measurements are plotted over the mean of the measurements, and $\bar{d}$ and the limits of agreement are added as horizontal lines, the resulting plot is called a Bland–Altman plot (sometimes also referred to as Tukey mean difference plot). It can be used to quickly visually inspect how well measurements can be reproduced, specifically which systematic bias ($\bar{d}\neq 0$) and which variability or “spread” of measurement differences to expect. Accordingly, we assessed the RC, limits of agreement, and produced Bland–Altman plots for both methods, staircase and QUEST, to gain more insight into the repeatability (or lack thereof) of measurements for each method. The use of these analyses requires the measurement differences to be normally distributed, which we confirmed using Q–Q plots, and Shapiro–Wilk tests failed to reject the null hypothesis of normal distributions (all p>0.05). Confidence intervals for the limits of agreement were calculated using the “exact paired” method [21].

Lastly, to test whether the duration of the inter-session interval might be a confounding factor in the threshold estimates, we also calculated the Spearman correlation between inter-session intervals and differences between Test and Retest thresholds.

##### Comparison between Procedures

To compare the threshold estimates across procedures, we averaged Test and Retest thresholds for each participant within a procedure, and, similarly to the analysis of reliability, compared the means with a Wilcoxon signed-rank test, followed by the calculation of Spearman’s ρ and the fit of a regression line using an OLS model. The regression residuals were normally distributed, according to a Q–Q plot and a Shapiro-Wilk test (W=0.96, p=0.26), satisfying the normality assumption of errors on which OLS regression crtitically relies.

Additionally, we estimated the 95% limits of agreement from the differences between the within-participant session means for the two procedures, and generated Bland-Altman plots. The measurement differences were normally distributed, according to a Q-Q plot and a Shapiro-Wilk test (W=0.96, p=0.30). Like in the investigation of test-retest reliability, we assessed confidence intervals of the limits of agreement via the “exact paired” method [21].

Because the limits of agreement derived from session means might actually be too narrow, as within-participant variability is removed by averaging measurements across sessions [20], we calculated adjusted limits of agreement from the variance of the between-subject differences, $\sigma_{d}^{2}$, which in turn can be calculated as $\sigma_{d}^{2}=s_{\bar{d}}^{2}+0.5\;s_{xw}^{2}+0.5\;s_{yw}^{2}$. Here, $s_{\bar{d}}^{2}$ is the variance of the differences between the session means; and $s_{xw}^{2}$ and $s_{yw}^{2}$ are the within-participant variances of methods *x* and *y*, respectively (staircase and QUEST in our case). The limits of agreement can then be calculated as $\bar{d}\pm 1.96\;\times \sigma_{d}$, with $\bar{d}$ being the mean difference between the session means of both procedures. Again, the interpretation of these limits is straightforward: 95% of the differences between staircase and QUEST measurements can be expected to fall into this interval, and narrower limits indicate a better agreement across the measurement results produced by both procedures. Finally, we derived 95% confidence intervals for these limits ([20], Section 5.1, Equation (5.10)).

##### Software

The experiments were run via PsychoPy 1.85.4 [22,23] running on Python 2.7.14 (https://www.python.org) installed via the Miniconda distribution (https://conda.io/miniconda.html) on Windows 7 (Microsoft Corp., Redmond, WA, USA). All analyses were carried out with Python 3.7.1, running on macOS 10.14.2 (Apple Inc., Cupertino, CA, USA). We used the following Python packages: correlation coefficients, Bland-Altman and Q-Q plots were derived via pingouin 0.2.2 [24]; confidence intervals for the Bland–Altman plots were calculated with pyCompare 1.2.3 (https://github.com/jaketmp/pyCompare); Shapiro–Wilk statistics were calculated with SciPy 1.2.1 [25,26]; linear regression models were estimated using statsmodels 0.9.0 [27]; and box plots and correlation plots were created with seaborn 0.9.0 (https://seaborn.pydata.org) and matplotlib 3.0.2 [28].

## 3. Results

### 3.1. Odor Discrimination and Identification

The average test score was 13.3 (SD = 1.5, range: 11–16; N=35) for odor discrimination, and 13.0 (SD = 1.6, range: 11–16; N=36) for odor identification. When summed with the staircase threshold estimates from the Test and Retest sessions, we observed TDI scores of 33.34 (SD = 3.8; range: 26.5–43) and 33.64 (SD = 3.8; range: 26.75–41.75), respectively. Individual as well as cumulative scores indicate a below-average ability to smell (roughly around the 25th percentile) in our sample compared to recent normative data from over 9000 subjects [8].

### 3.2. Starting Concentrations

The average starting concentration was pen no. 9.9 (SD = 4.2, range: 1–16) for the Test and 9.6 (SD = 4.1, range: 1–16) for the Retest session of the staircase. The average difference in starting concentrations between sessions was 4.9 (SD = 4.0, range: 0–15). In comparison, we used a slightly higher, fixed starting concentration of pen no. 7 for QUEST.

### 3.3. Test Duration

The average number of trials needed to complete the staircase measurements was 23.6 (SD = 4.8, range: 13–41), which translates to approx. 11.5 min and is 2 minutes longer than for QUEST, which per our parameters always lasted 9.5 minutes (20 trials). Test duration varied slightly between staircase sessions and was 24.4 trials (SD = 4.2, range: 16–34) for the test and 22.9 trials (SD = 5.4, range: 13–41) for the retest session. Please note that the number of trials and the testing duration for the staircase are based on the time required to reach seven reversal points after the starting concentration had been determined, thereby deviating from the “standard” procedure, which treats the starting concentration as the first reversal.

### 3.4. Test-Retest Reliability

The mean Test thresholds did not differ from the mean Retest thresholds for the staircase ($M_{\mathrm{Test}}=6.9$, ${\mathrm{SD}}_{\mathrm{Test}}=3.1$; $M_{\mathrm{Retest}}=7.2$, ${\mathrm{SD}}_{\mathrm{Retest}}=3.2$; W=268.0, p=0.19). For QUEST, on the other hand, mean test and retest thresholds differed significantly, with slightly higher sensitivity (higher *T* unit) in the Retest ($M_{\mathrm{Test}}=5.2$, ${\mathrm{SD}}_{\mathrm{Test}}=3.8$; $M_{\mathrm{Retest}}=6.2$, ${\mathrm{SD}}_{\mathrm{Retest}}=3.4$; W=201.5, p<0.01; see Figure 1).

The test and retest thresholds correlated significantly for both procedures, with QUEST demonstrating a stronger relationship between measurements than the staircase (staircase: $\rho_{34}=0.49$, p<0.01; QUEST: $\rho_{34}=0.66$, p<0.001; Figure 2A).

As already pointed out, correlation gives an indication of the strength of the monotonic relationship between values, but only provides limited information on their agreement. We therefore calculated the repeatability coefficient RC and created Bland–Altman plots to generate a better understanding of the measurement differences. The prediction of the RC is that two measurements (test and retest) will differ by the value of RC or less for 95% of participants. We found that RC was about 16% smaller for QUEST than for the staircase (${\mathrm{RC}}_{\mathrm{Staircase}}=6.44$, ${\mathrm{RC}}_{\mathrm{QUEST}}=5.43$), suggesting a slightly better agreement between Test and Retest measurements for the QUEST procedure. Accordingly, the Bland–Altman plot (Figure 2B) showed narrower limits of agreement for QUEST (staircase: -6.79[-8.89,-5.63] and 6.09[4.93,8.18]; QUEST: -6.42[-8.18,-5.44] and 4.44[3.46,6.29]; 95% CIs in brackets). The mean of the differences between measurements was relatively small and deviated less than 1 *T* unit from zero—the “ideal” difference—for both methods ($M_{\Delta T,\mathrm{Staircase}}=-0.35$[-1.43,0.72]; $M_{\Delta T,\mathrm{QUEST}}=-0.99$[-1.89,-0.08]). This systematic negative shift indicates that participants, on average, reached higher *T* units in the second session than in the first. The differences between Test and Retest measurements for three (staircase) and two participants (QUEST), respectively, fell outside their respective limits of agreement, which corresponds to the expected proportion of 5% of outliers (336=8.3%; 236=5.6%), demonstrating the appropriateness of the estimated limits. Considering the confidence intervals of the limits of agreement, an equal number of measurement differences (four) fell outside the predicted range for both procedures.

To test whether the time between Test and Retest sessions might be linked to the observed differences between Test and Retest threshold estimates, we computed correlations between those measures. We found no relationship for either method (staircase: $\rho_{34}=-0.12$, p=0.50; QUEST: $\rho_{34}=0.03$, p=0.85).

### 3.5. Comparison between Procedures

Although the threshold estimates, averaged across sessions, for the staircase were significantly higher than those for QUEST ($M_{\mathrm{staircase}}=7.0$, ${\mathrm{SD}}_{\mathrm{staircase}}=2.7$; $M_{\mathrm{QUEST}}=5.7$, ${\mathrm{SD}}_{\mathrm{QUEST}}=3.3$; W=101.0, p<0.001; Figure 3A), we found a strong correlation between the procedures ($\rho_{34}=0.80,$p<0.001; Figure 3B). The regression slope was close to 1, providing an indication of agreement across procedures. The Bland-Altman plot based on the session means (Figure 3C) shows a systematic difference between both procedures; specifically, QUEST thresholds were, on average, 1.38[0.78,1.97]*T* units smaller than the staircase estimates (95% CIs in brackets). The limits of agreement reached from -2.20[-3.37,-1.56] to 4.95[4.31,6.12], meaning the difference between the two procedures will fall into this range for 95% of measurements. Only for 1 participant the observed differences between staircase and QUEST fell outside the limits of agreement (136=2.8%; when considering the CIs of the limits, 3 participants fell outside the expected range (336=8.3%)

The corrected limits of agreement, taking into account individual measurements (as opposed to session means only), were -4.20[-23.6,15.3] and 6.96[-12.5,26.4], which is substantially larger than the uncorrected limits. The large confidence intervals that expand even beyond the concentration range reflect the relatively large within-participant variability across sessions in both threshold procedures.

## 4. Discussion

In the presented study we used a QUEST-based algorithm to estimate olfactory detection thresholds for 2-phenylethanol with the aim to provide a reliable test result as it had recently been demonstrated for taste thresholds [13] with reduced testing time. The results were compared to a slightly modified version of the widely-used testing protocol based on a one-up/two-down staircase procedure [6,7,9,15,16].

Test–retest reliability was assessed using multiple approaches. Comparison of Test and Retest thresholds revealed a small yet significant mean difference for QUEST: threshold estimates during retest were higher than in the test, indicating an increase in participants’ sensitivity. A similar effect was reported in a previous study [6]. However, with a mean difference of approx. 1 *T* unit or pen number, the practical relevance of this effect is debatable, even more so when considering the large variability of measurement results within individual participants.

Following common practice of establishing test-retest reliability of olfactory thresholds (see e.g., [6,9,29]), we calculated correlations between Test and retest sessions. The correlation coefficient for QUEST (ρ=0.66) indicated solid, but not exceptionally great test–retest reliability. Reliability of the staircase procedure was only moderate (ρ=0.49) and lower than reported in previous studies for *n*-butanol (r=0.61; [6]) and 2-phenylethanol (r=0.92; [9]) thresholds.

To acknowledge previous criticism of correlation analysis – which focuses on the agreement, but not on the differences between measurements [18,19,20] – we calculated repeatability coefficients and generated Bland–Altman plots for the analysis of session differences. Repeatability was higher for QUEST than for the staircase; however, measurement results of both procedures varied considerably across sessions for many participants. This inter-session variability is further substantiated by the differences in starting concentrations assessed for the staircase, which varied up 15 pen numbers in the most extreme case. The effect was not universal: some participants performed better in the Test than in the Retest session, whereas for others performance dropped across sessions, and remained almost unchanged in others. Since both sessions had been scheduled within a relatively short time period and all measurements have been performed by the same experimenter, measurement variability can be mostly attributed to variability within participants themselves.

The comparison of the staircase and QUEST procedures via the session means of each participant showed that the staircase yielded slightly higher pen numbers (i.e., lower thresholds) than QUEST. This was expected as the procedures were assumed to converge at approx. 71% and 80% correct responses, respectively. We found a strong correlation between the session means of the procedures (ρ=0.80), and regression analysis showed an almost perfect linear relationship, which some would interpret as a good agreement between QUEST and staircase results. The 95% limits of agreement, taking into account the within-participant variability, showed a large expected deviation between both procedures (range: QUEST thresholds almost 7 *T* units smaller or more than 4 *T* units greater than staircase results), with the corresponding CIs of those boundaries even exceeding the concentration range. This result is indicative of the large variability we found within participants in both procedure. The limits of agreement based on the within-participant session means were much narrower, as variability is greatly reduced through averaging.

A potential source of variability might be guessing. In fact, the probability of responding correctly merely by guessing is 13.

In a series of simulations, it could be shown that with an increasing number of trials the frequency of correct guesses might get unacceptably high, potentially leading increased variability in the threshold estimates [30]. The author determined that, for a staircase procedure like the one in our study, the expected proportion of such false-positive responses exceeds 5% with the 23rd trial. For our staircase experiments, the average number of trials was 23.6; and the procedure finished after 23 or more trials for 24 of the 36 participants in the Test, and for 20 participants in the Retest session. Therefore, the large variability between Test and Retest threshold estimates in the staircase could, at least partially, be ascribed to correct guesses “contaminating” the procedure. However, QUEST—which always finished after 20 trials—only had slightly better test-retest reliability according the the repeatability coefficient, suggesting that the largest portion of test-retest variability in our investigations was probably not caused by (too) long trial sequences and related false-positive responses alone.

Surprisingly, a number of participants were unable to correctly identify pen no. 1 at least on one occasion, and this effect was more pronounced during QUEST compared to the staircase. It seems plausible that the variable step size used by QUEST made it possible to approach even the extreme concentration ranges quickly, whereas the staircase requires a longer sequence of incorrect responses to reach pen no. 1.

Despite careful selection of healthy participants who reported no smell impairment, olfactory performance was lower than recently reported in a sample comprising over 9000 participants [8]. This coincidental finding highlights the need for a comprehensive smell screening before enrollment. To what extend olfactory function contributed to the present results and limits their generalizability remains to be explored.

All QUEST runs completed after 20 trials for all participants. The procedure could be further optimized by introducing a dynamic stopping rule. For example, [13] set the algorithm to terminate once the threshold estimate had reached a certain degree of confidence. Such a rule can reduce testing time, as the run may finish in fewer than 20 trials, and should be considered in future studies. Although the reduction or omission of a minimum trial number bears potential to reduce the testing time further, it needs to be shown first that the algorithm performs well under these conditions and, most importantly, large-scale studies need to show whether such a reduced or faster protocol is appropriate to assess odor sensitivity in participants with odor abilities at the extremes (particularly insensitive/sensitive).

Inspection of the data showed that some staircase runs had not fully converged although seven reversal points were reached. In these cases, participants exhibited a somewhat “fluctuating” response behavior (or threshold) that caused the procedure to move in the direction of higher concentrations throughout the experiment (see Figure A1 in the Appendix A and supplementary data for an example). QUEST proved to behave more consistently, at least in some cases, by either converging to a threshold or by reaching pen no. 1, which would then sometimes not be identified correctly. These interesting differences between procedures require further investigation to fully understand their cause and influence on threshold estimates and, ultimately, diagnostics.

## 5. Conclusions

The present study compared the reliability of olfactory threshold estimates using two different algorithms: a one-up/two-down staircase and a QUEST-based procedure. The measurement results of both procedures showed considerable overlap. QUEST thresholds were more stable across sessions than the staircase, as indicated by a smaller variability of test-retest differences and a higher correlation between session estimates. QUEST offered a slightly reduced testing time, which may be further minimized through a variable stopping criterion. Yet, QUEST also tended to present the highest concentration, pen no. 1, more quickly than the staircase, which may induce more rapid adaptation and habituation during the procedure and, eventually, produce biased results. Further research is needed to better understand possible advantages and drawbacks of the QUEST procedure compared to the staircase testing protocol.

## 6. Data and Software Availability

The data analyzed in this paper along with graphical representations of each individual threshold run are available from https://doi.org/10.5281/zenodo.2548620. The authors provide a hosted service for running the presented experiments online at https://sensory-testing.org; the sources of this online implementation can be retrieved from https://github.com/hoechenberger/webtaste.

## Figures and Tables

**Figure 1 nutrients-11-01278-f001:**
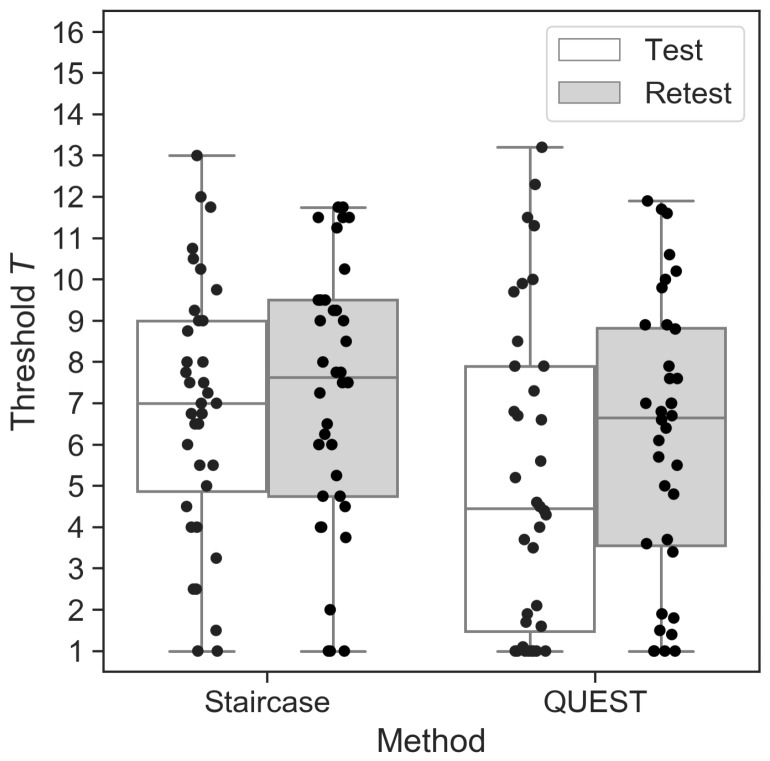
Threshold estimates for the staircase and QUEST procedures during Test and Retest sessions. Each dot represents one participant. Horizontal lines show the median values, and whisker lengths represent 1.5× inter-quartile range.

**Figure 2 nutrients-11-01278-f002:**
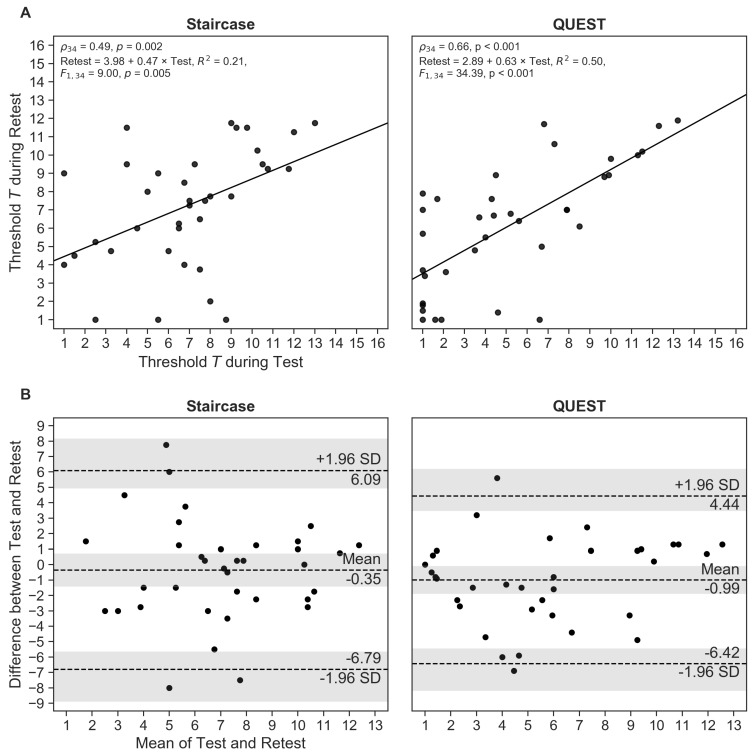
(**A**) Correlation between Test and Retest threshold estimates for the staircase and QUEST procedures. (**B**) Bland–Altman plots showing mean differences between Test and Retest, and limits of agreement corresponding to 95% confidence intervals (CIs) as mean±1.96×SD. The shaded areas represent the 95% CIs of the mean and the limits of agreement. Each dot represents one participant.

**Figure 3 nutrients-11-01278-f003:**
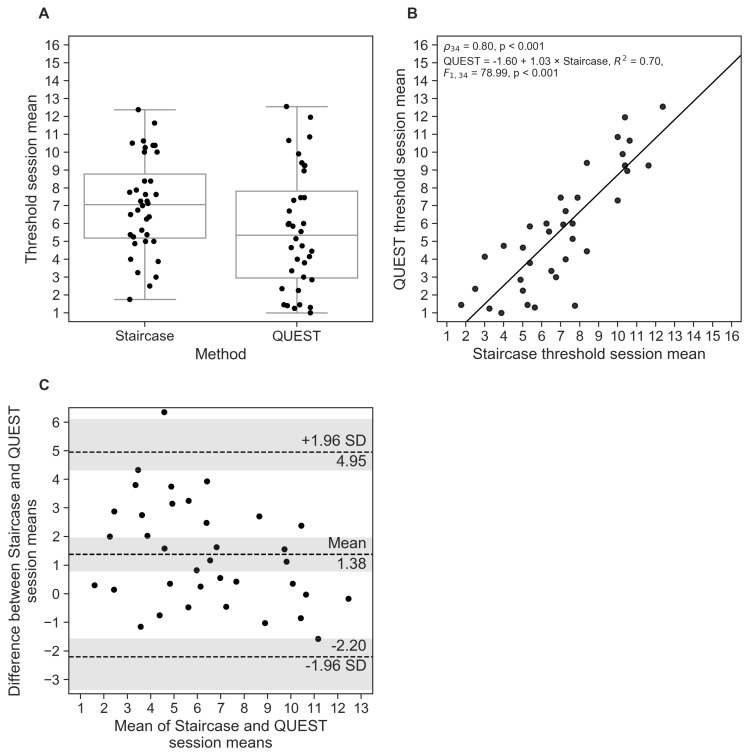
(**A**) Mean threshold estimates, averaged across Test and Retest sessions for the staircase and QUEST procedures. Horizontal lines show the median values, and whisker lengths represent 1.5× inter-quartile range. (**B**) Correlation between mean staircase and QUEST threshold estimates. (**C**) Bland–Altman plot showing mean differences between session means in both procedures, and limits of agreement corresponding to 95% confidence intervals (CIs) as mean±1.96×SD. The shaded areas represent the 95% CIs of the mean and the limits of agreement. Each dot represents one participant.

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
