# Peer review of "Estimation of Olfactory Sensitivity Using a Bayesian Adaptive Method"

_nutrients, 2019, doi:10.3390/nu11061278_

Round 1

Reviewer 1 Report

The  manuscript has been improved according to the suggestions of reviewers. No further changes required. 

Author Response

Thank you for the positive ealuation.

Reviewer 2 Report

I know that the current version is written in response to reviewer comments, but it's currently pretty onerous to read based on the extensive data analysis and results sections.  I think that some of the information could be moved to the supplementary section so that your overall findings are better highlighted.  

Author Response

Thank you for the positive evaluation of our manuscript.

Although we can understand the suggestion to move parts of the methods and results to the supplement, we fear that if we moved entire paragraphs to the supplement, the readers would not fully comprehend the rationale of the analyses employed here and we hence opted to leave them in the main text. 

Reviewer 3 Report

While I appreciate the authors clarifications, I still have fundamental concerns about the methods and validity of the data reported in this manuscript. The authors have substantially upped the statistical verbiage of the article, but do not address the fundamental underlying issues at hand in a practical way. The amount of variability and the way that variability is handled in the data analysis should really make the authors question the methods, yet the discussion does not seem to focus on these practical issues. 

As another of the reviewers also pointed out, there are methods to account for repeated measures that would allow the authors to not average out the variability in test, retest. Finding no difference between test, retest is not a good argument, as with great variability you would not get differences. That's the entire point of including all the measurements, rather than averaging away the variation for the sake of simpler (but inaccurate) statistics. 

The authors are correct in their response that the residuals, not the raw data, need to be normal. I brought this up because in that paragraph the authors state the data were not normal... if for the least squares analysis, the residuals are normal (though the raw data were not) please do include that in the text so that the reader understands.  

I don't understand the new text in the data cleaning section. This needs to be extremely clear, as it greatly influences the results. Did the authors, or did they not, repeatedly use pen 1 until a correct response occurred? What is the second paragraph saying? How many people could not consistently ID pen 1? How many times was it tried?

The fact that your sample appears to have below average ability is of great concern. Especially in the QUEST method, there are a LOT of people at pen 1 in figure 1. The usefulness of all of this work in unfortunately called in to question because how do you know it was the method or the subjects? 

The authors discuss the staircase failing to converge, which again seems of critical importance in any conclusions. Adding more statistics does not solve this problem. More participants should be tested, or other modifications of the staircase method considered, if the authors actually want to compare approaches. Reporting a "standard" method that did not seem to work "standardly" is a pretty big concern if trying to show how a new approach compares. 

Figure 2 also shows the inconsistency in these methods, which should be of high concern. Having the 95% CI include a total of 10-12 steps, when there are only 16 pens, should cast some doubt on the reliability of this from a very practical standpoint. Even if the statistics give numbers that could be claimed to be "reliable"--really, look at the data. Stats are only a tool. The data shows in these figures indicate there is an enormous amount of noise. 

I suspect QUEST shows a stronger relationship because of all the "pen 1" conclusions... especially that cluster of 5 or so subjects at Pen 1 and 2 consistently. 

Author Response

While I appreciate the authors clarifications, I still have fundamental concerns about the methods and validity of the data reported in this manuscript. The authors have substantially upped the statistical verbiage of the article, but do not address the fundamental underlying issues at hand in a practical way. The amount of variability and the way that variability is handled in the data analysis should really make the authors question the methods, yet the discussion does not seem to focus on these practical issues.

As another of the reviewers also pointed out, there are methods to account for repeated measures that would allow the authors to not average out the variability in test, retest. Finding no difference between test, retest is not a good argument, as with great variability you would not get differences. That's the entire point of including all the measurements, rather than averaging away the variation for the sake of simpler (but inaccurate) statistics.

We agree with the reviewer that averaging multiple measurements reduces variability. This can be both an appreciated and an undesired effect, depending on the specific application under consideration.

For example, assuming a sufficiently reliable method, averaging multiple measurements can indeed reduce “noise” and produce a better parameter estimate. Given the only mediocre repeatability esp. of the staircase in our study, however, one can indeed question this approach. This is precisely why we included an analysis that takes into consideration all measurements without averaging; this is described in detail in the Methods section under “Comparison between procedures” (page 6). There we state, “[because]  the  limits  of  agreement  derived  from  session  means  might  actually  be  too  narrow, as within-participant variability is removed by averaging measurements across sessions, we calculated adjusted limits of agreement from the variance of the between-subject differences”.

The authors are correct in their response that the residuals, not the raw data, need to be normal. I brought this up because in that paragraph the authors state the data were not normal... if for the least squares analysis, the residuals are normal (though the raw data were not) please do include that in the text so that the reader understands.

Reply: We have added the information to text that residuals were normally distributed according to Q-Q plots and Shapiro-Wilk tests.

I don't understand the new text in the data cleaning section. This needs to be extremely clear, as it greatly influences the results. Did the authors, or did they not, repeatedly use pen 1 until a correct response occurred? What is the second paragraph saying? How many people could not consistently ID pen 1? How many times was it tried?

Reply: We agree with the reviewer that it is important to be very explicit about any treatment of the raw data. For this reason, we have reported in how many cases subjects failed to identify the highest concentration because these runs were assigned a threshold value of 1 (see paragraph Data Cleaning). This does not mean, however, that participants never identified pen no. 1 (or other pens) correctly or that we aborted the threshold measurement. The staircase always ended after 7 reversals; QUEST ended after 20 trials. For full transparency, we had also uploaded the raw data as well as a graphical summary of all individual thresholds runs thereby allowing the reader a comprehensive review.

In our previous reply to reviewer no.2, we had described the response behavior of those subjects who failed to identify pen no 1. at least once in more detail. We simply assumed that all reviewers would see our replies to the other reviewer. If this was not the case, we apologize that we did not detail this for the reviewer as well.

For the STAIRCASE, we report in the paper that 5 participants failed to identify pen no. 1 at least once. All of these participants correctly identified several lower concentrations within the same threshold run: two participants correctly identified stimuli up to pen no. 4 , one identified up to no. 5,  one identified even concentrations up to pen 10 and yet another one up to pen no. 14, which represents an extremely low concentration.

For QUEST 11 participants failed to identify pen no. 1 at least once. In all but one of these participants pen no. 1 and also less concentrated pens were identified on other trials during the same threshold measurement - in these cases pens no. 2, 4, 5, 7, and 8 were correctly identified twice and no. 3 once.  

The fact that your sample appears to have below average ability is of great concern. Especially in the QUEST method, there are a LOT of people at pen 1 in figure 1. The usefulness of all of this work in unfortunately called in to question because how do you know it was the method or the subjects?

Reply: To emphasize that our sample exhibited a below-average ability to smell when compared to the most recent norm-data, we have included a section in the discussion (in addition to the description in the methods/results), where we state: “Despite careful selection of healthy participants who reported no smell impairment, olfactory performance was lower than recently reported in a sample comprising over 9,000 participants [8]. This coincidental finding highlights the need for a comprehensive smell screening before enrollment. To what extend olfactory function contributed to the present results and limits their generalizability remains to be explored.”

The authors discuss the staircase failing to converge, which again seems of critical importance in any conclusions. Adding more statistics does not solve this problem. More participants should be tested, or other modifications of the staircase method considered, if the authors actually want to compare approaches. Reporting a "standard" method that did not seem to work "standardly" is a pretty big concern if trying to show how a new approach compares.

Reply: We agree with the reviewer that it is critical that the staircase seemed not to have fully converged in a few cases and also that this needs further investigation. Notably, we included a paragraph in the discussion (and a figure in the appendix) to illustrate this important observation and to highlight the need for further research on the matter. As far as we can tell, convergence after 7 reversals had been implicitly assumed in previous work. We found it therefore particularly important to report our finding and to encourage that other researchers be aware of this potential issue.

Figure 2 also shows the inconsistency in these methods, which should be of high concern. Having the 95% CI include a total of 10-12 steps, when there are only 16 pens, should cast some doubt on the reliability of this from a very practical standpoint. Even if the statistics give numbers that could be claimed to be "reliable"--really, look at the data. Stats are only a tool. The data shows in these figures indicate there is an enormous amount of noise.

Reply: The variability of the data is indeed quite large. Whether the “noise” seen here is due to the measurement tools themselves or caused by other factors in unknown. We selected multiple analysis approaches and we display the data in various ways  in an attempt to illustrate different aspects and facets of the results. Nowhere do we claim that any of the methods is working particularly reliably, nor are we basing our interpretations solely on numbers. Additionally to the statistics and figures included in the paper, we provide plots of the trial sequences for every single measurement and the dataset we based our analyses upon online, allowing the interested reader to get an even more detailed impression of the data, should they desire to do so. The staircase method we applied was based on an established diagnostic tool; we therefore, too, were surprised it didn’t work too well in our specific investigation.

I suspect QUEST shows a stronger relationship because of all the "pen 1" conclusions... especially that cluster of 5 or so subjects at Pen 1 and 2 consistently.

Reply: The reviewer draws a very intuitive conclusion. However, when considering the data, once can see that only a single participant had a threshold of 1 during QUEST test as well as during QUEST retest. All other participants who exhibited a threshold of 1 in either of the two sessions had a different threshold during the corresponding session: the differences between sessions was quite substantial and ranged from 0.5 to 6.9 pens. It hence seems that participants who had a threshold of 1 contributed as much to variability (and decorrelation) as they support the correlation.

This manuscript is a resubmission of an earlier submission. The following is a list of the peer review reports and author responses from that submission.

Round 1

Reviewer 1 Report

The manuscript presents an interesting study, but is inherently flawed because the order of the stimuli wasn’t randomized for one of the methods, but was for the other. Participants could have guessed the order. I am very sorry for the authors, but that is critical. If the experiment is re-done with randomized order of the stimuli, then it would be very interesting. As it stands, there is no way to assess whether these results are valid because of this flaw.

Some other aspects of the analysis could be improved, but this major flaw makes it difficult to evaluate the legitimacy of any of the results; detailed comments below. 

Line 20: bitter “note”—please be precise. Do you mean taste?

Line 62: “to” = “two”?

Line 96: Do you mean 20s total for all 3, or 20s in between each pen?

Line 98 – 100: This type of set order is really biased, and could easily be figured out by a participant. The order should be randomized. The citation given used randomized order, not a set order like the authors describe.  Please clarify if I have misinterpreted; this is absolutely critical, and could invalidate the results. As you mention it again in line 116 and 136, I don’t think I have misinterpreted.

Regarding staircase method in general: More recent work has shown with random number generators that the results of a 1 up 2 down staircase methods can actually have really high false positive rates. This seems a critical issue with these detection threshold measures. Please consult that work (doi: 10.3758/s13414-014-0798-9) and compare to your observations.

Lines 125+: I’m concerned about the potentially large difference in starting concentration from one method to another. Give the above mentioned false positives, this could really bias the outcomes. At very least, data on the starting concentrations for the participants should be shown and application of potential false-positives rates considered.

Line 150: A value outside the range would be more appropriate (i.e., 0.5 or 16.5); this is generally the approach when using actual concentrations, rather than integer representations, as explained in various ASTM methods for thresholds.

Line 159: Entirely dropping the non-converging participants, and making these decisions visually, could inherently bias the data. Rules need to be developed beforehand. The non-convergence implies the method does not work, which is a critical consideration.

Line 174: Please explain why OLS was used if the data were not normally distributed—doesn’t this use the same assumptions as any linear regression, so a violation of normality, as already described, would not be acceptable?

Line 181: These outlier sessions are important. Repeated measures approaches should be used to keep these values in the dataset, rather than averaging out the variability, which is critical to understanding actually differences (or lack thereof because of excess variability) in the methods.

Line 239: This argument does not make sense. If there is no gender effect in larger population, then how could there be in a small one?

Reviewer 2 Report

This is an important study for researchers examining olfactory thresholds.  However, one of the strongest arguments for using a method like QUEST rather than a trusted and established staircase method is the potential for QUEST to determine threshold more quickly than a staircase.  The differences in testing time (if there were any) are not explicitly addressed in the manuscript, which I think is a significant oversight.  I am also concerned that the sample was so unbalanced between men and women.  This could be addressed by removing the men from the sample and rerunning analyses, and/or explicitly stating whether there were any outliers in the data.  Finally, I think that the high frequency of identification failure for pen no1, especially in the QUEST procedure, is concerning.  There is no expected frequency for anosmia to PEA, and therefore the fact that 30% of your participants could not identify pen 1 in the QUEST method makes me question whether there was something wrong with your stimuli - perhaps that the pens were old or not sufficiently saturated with PEA, or that your experimenter held the pens too far from the noses of your participants.  You do not elaborate on how many of these participants correctly identified pens at midrange concentrations. Further, though you have collected the data, you do not report whether or not these participants displayed normosmic function in the other olfactory tests conducted.  I'm sure you are working on a second publication, but it might be worth a quick mention in order to clarify the unexpectedly large number of participants who could not identify pen 1.  A plausible explanation should be provided. 

As a side note, there are several typos in the manuscript, so please reread carefully for grammatical issues (see, e.g., lines 35, 62, 161).

Overall, I think the work presented in this paper is valuable, but some items need to be clarified before publication to maximize the impact of these findings.

Reviewer 3 Report

The manuscript presents a significant research study to improve the Sniffin’ Sticks test method. I suggest publication of the manuscript after consideration of some comments below.

Overall Comments:

·       Data Analysis: Have you tried Repeated Measure (RM) ANCOVA method to analyze your data? Your data points were collected over two sessions with the same participants. That said, the participants’ ages can be a covariate as well as the time between the sessions because these could be related to the odor sensitivity. This would be a better way to analyze your threshold data that fluctuate across session [Line 180].

·       Please include averaged testing time taken for both of QUEST and staircase procedures to make it clear that the QUEST would require less time to complete than the staircase procedure.

Abstract Line 3 & Line 278-279: Could you please give some examples of the ‘some cases’ that QUEST would work better?

Line 60: Were there any exclusion criteria such as heavy smokers? The number of participants who were unable to identify correctly pen no. 1 seems pretty high even with the staircase procedure based on my experience with sniffin sticks (Line 195-197). Thus, I was wondering what demographic profile of your participants looks like other than the age range. 

Line 80: Please include an average time between the two sessions for all of the participants.

Line 83-85: Why odor identification and discrimination test were measured? Is it before or after the two ‘thresholds’ tests (Quest and staircase)? 

Line 146 & Line 266-267: How many trials were usually needed for 7 reversal points staircase procedure?

Line 241-244: The better performance in the second session might be due to learning effects? Would you recommend researchers plan on having two sessions if they want to use QUEST procedure because the first and second sessions are significantly different? If so, the threshold should be averaged across sessions or just the threshold from second session? Please discuss further regarding why it was shown to have significantly lower threshold in the second session and what it means to the researchers who want to use the QUEST procedure.